# Study of Factors Affecting the Copper Ore Leaching Process

**Aigul Koizhanova, Bagdaulet Kenzhaliyev, David Magomedov \*** , **Emil Kamalov, Mariya Yerdenova, Akbota Bakrayeva and Nurgali Abdyldayev**

JSC Institute of Metallurgy and Ore Beneficiation, Satbayev University, Shevchenko 29/133, Almaty 050010, Kazakhstan

\* Correspondence: davidmag16@mail.ru

**Abstract:** This paper provides an overview of hydrometallurgical copper extraction studies in which liquid extraction technology has been used with four copper deposits of different compositions. The sulfuric acid consumption rate and copper extraction efficiency, which are dependent on the initial content and forms of calcium compounds and other impurities in ore samples, were calculated, and the results are presented herein. It was established that during the leaching process, silicate compounds of alkaline earth metals, in addition to calcium and magnesium carbonate compounds, would affect the levels of sulfuric acid consumption, thereby actively lowering the acidity of the environment. Moreover, these compounds could partially sorb copper ions from sulfuric acid leaching solutions. Thus, the analysis of waste ore samples showed that residual copper is mainly contained in the form of complex silicate complexes. The presence of divalent iron compounds in the composition from one of the deposits also allowed us to perform a biochemical leaching experiment with preliminary oxidation using an *Acidithiobacillus ferrooxidans* bacterial culture adapted to the ore composition. The use of this biochemical method in the copper leaching process resulted in a significant reduction in sulfuric acid consumption, by 40%, and a copper recovery rate of 87.2%.

**Keywords:** copper leaching; sulfuric acid consumption; acid-intensive minerals; bio-oxidation

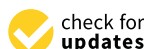



## 1. Introduction

Copper deposits currently involved in hydrometallurgical processing tend to have high amounts of impurities, which complicate the sulfuric acid leaching process. In the 1970–90s, it was suggested that different additives be used to intensify the sulfide copper ore leaching process with acid solutions, such as sulfuric and nitric iron (III) and ammonium salts, fluoride ions, surface-active substances, oxygen, ozone, sodium chloride, nitrates and chlorides of alkaline, and alkaline earth metals [1–6], which accelerate the dissolution process and decrease sulfuric acid consumption. Leaching methods for copper mineral raw materials with complex compositions at high pressures, temperatures, and concentrations of sulfuric acid have recently been studied [7,8], and these have enabled more than 90% of copper to be extracted. The extraction processes of the selective collection of various groups of metals via cationic and anionic type liquids are widely represented in the works of modern researchers [9–11]. The practical applicability and economic feasibility of innovative methods for liquid extraction production technology are important factors, in addition to increased rates of copper extraction. Thus, the presence of nitrate and chloride ions in productive copper solutions will adversely affect the subsequent stages of extraction and re-extraction after the use of appropriate salts as additives to accelerate the leaching process. Alternative solutions for the copper leaching process, which are often considered for ores with high levels of acid-intensive minerals, are also unacceptable for liquid extraction technology or are economically unprofitable. The most used acid-absorbing minerals are calcium compounds, often calcites and dolomites. In some cases, calcite in combination with certain carbon compounds can act as a sorbent for copper ions in sulfuric

acid solution. This property is disclosed in [12], where the adsorption mechanism responsible for the removal of $Cu^{2+}$ with final precipitation in the form of the mineral posniakite $(Cu_4[(OH)_6SO_4] \cdot H_2O)$ is described.

Beneficiation methods for the flotation of ores of similar composition are used when copper is mainly in the sulfide form; further possibilities are to use pyrometallurgical melting methods, such as in the example of manganese ores with high calcium content [13], or vacuum separation in the example of ores of rare-earth metals [14]. Liquid extraction technology of the sulfuric acid leaching method remains the main processing method for off-balance ores and waste dumps which predominantly comprise oxidized forms of copper. Therefore, the study of the effect of the composition of copper dump ores on the copper recovery rate and the final consumption of sulfuric acid, as well as the search for ways to improve the profitability of processing of off-balance ores and waste dumps, are important scientific and production tasks [15].

The data obtained during the corresponding experiments conducted at the Institute of Metallurgy and Ore Beneficiation JSC were reviewed here and analyzed to identify and describe the regularities of increases in the sulfuric acid consumption level during the copper leaching process. This article includes the results of copper leaching experiments in the full hydrometallurgical cycle of four copper dumps from different deposits located in Central Kazakhstan (Figure 1). Thus, data on the initial composition of ore material and sulfuric acid leaching results were obtained from the study projects of the following deposits, conducted during the indicated period: 2014–2016, the Bayskoye deposit; 2015, the Baitemir deposit; 2016–2018, the Sayak deposit; and 2021–2023, Satbayev town copper dumps, including bioleaching experiments. The chemical and phase compositions of ore material were analyzed, the optimum conditions for the leaching process were selected, the copper extraction efficiency was assessed, and the sulfuric acid consumption was recalculated per ton of ore and based on the quantity of copper extracted from the solution that was determined when all the specified projects had been performed [16,17].

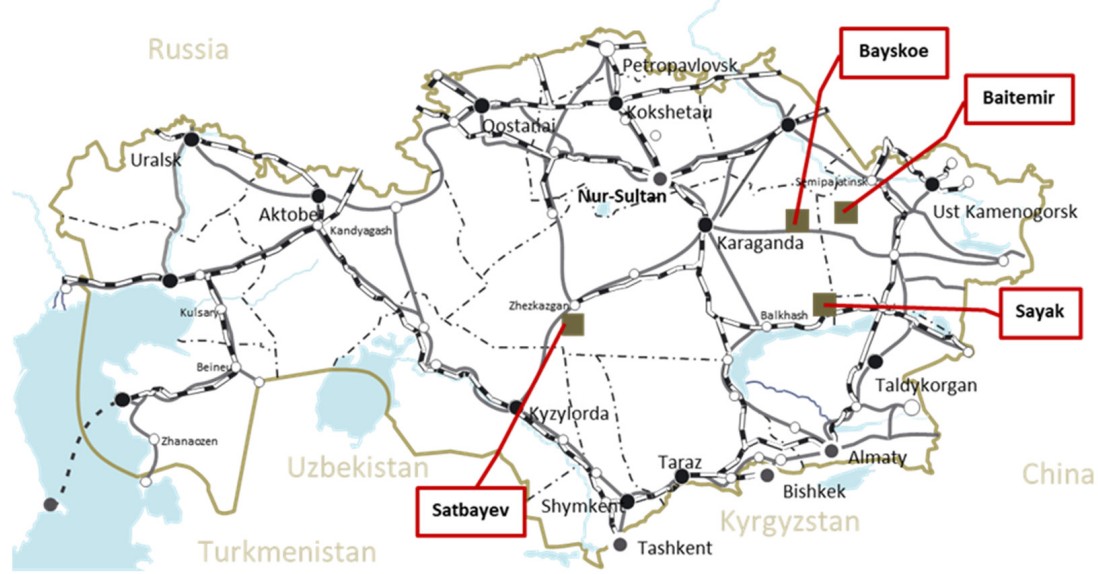

**Figure 1.** Geographical location of the studied deposits of copper dumps.

## 2. Study Subjects

Each deposit was sampled from different dump areas, and the ore material was then homogenized. Certain external differences were observed in the studied samples of dump ores. The samples from the Bayskoye and Satbayev deposits were predominantly formed of sandstone rocks, and clay fragments characterized the samples from the Sayak and Baitemir deposits (Figure 2).

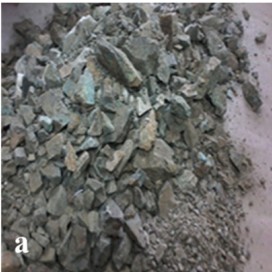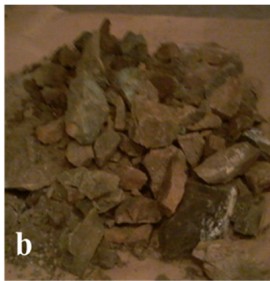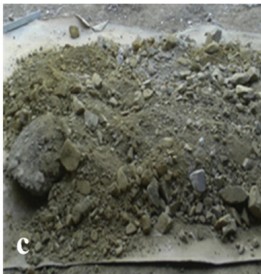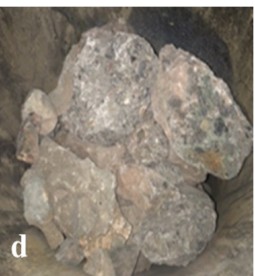

**Figure 2.** Appearance of samples taken from copper ore deposits of (**a**) Bayskoye; (**b**) Baitemir; (**c**) Sayak; and (**d**) Satbayev town dump.

The composition of copper ore samples was analyzed with the use of fluorescence and chemical methods. Table 1 shows the content of the main valuable component—copper—as well as metals with the highest content that affect the leaching process.

**Table 1.** Amount of copper and other metals in copper ore samples (%).

| Metal | Bayskoye | Baitemir | Sayak | Satbayev |
|---|---|---|---|---|
| Cu | 0.377 | 0.64 | 0.24 | 0.264 |
| Ca | 4.5 | 5.7 | 9.4 | 2.7 |
| Mg | 0.5 | 1.78 | 0.76 | 1.1 |
| Fe | 2.3 | 6.9 | 7.33 | 2.58 |
| Al | 7.13 | 5.1 | 5.1 | 5.8 |

From fluorescence analysis, significant percentages of silicon and oxygen were found in all samples, which is characteristic of quartz and silicate compounds, in addition to the metals specified in Table 1. Thus, the predominance of quartz and silicate compounds and other rock-forming fragments was established in all samples through subsequent X-ray phase analysis. The phases of copper minerals were clearly fixed only in the sample from the Satbayev dump ore, and no phases corresponding to copper compounds were found in the waste rock minerals in the remaining samples. The detailed X-ray phase analysis of all four samples is presented in Table 2.

**Table 2.** Results of X-ray phase analyses of the studied deposits.

| Bayskoye | | |
|---|---|---|
| **Compound Name** | **Formula** | **S-Q** |
| Albite | $Na(AlSi_3O_8)$ | 24.1 |
| Quartz | $SiO_2$ | 21.2 |
| Orthoclase | $K(Al.Fe)Si_2O_8$ | 18.4 |
| Clinochlore | $Al_2Mg_5Si_3O_{10}(OH)_8$ | 8.6 |
| Gypsum | $CaSO_4 \cdot 2H_2O$ | 6.9 |
| Tremolite | $(Ca \cdot Na \cdot Fe)_2Mg_5Si_8O_{22}(OH)_2$ | 6.6 |
| Muscovite-1M | $KAl_2Si_3AlO_{10}(OH)_2$ | 5.6 |
| Kaolinite | $H_4Al_2Si_2O_9$ | 4.0 |
| Laumontite | $Ca(Al_2Si_4O_{12}) \cdot 4H_2O$ | 3.1 |
| Calcite | $CaCO_3$ | 1.5 |
| **Baitemir** | | |
| **Compound Name** | **Formula** | **S-Q** |
| Quartz | $SiO_2$ | 64.2 |
| Montmorillonite. calcian | $(Ca \cdot Na)_{0.3}Al_2(Si \cdot Al)_4O_{10}(OH)_2 \cdot xH_2O$ | 11.8 |
| Bassanite. syn | $Ca(SO_4)(H_2O)_{0.662}$ | 10.6 |
| Muscovite | $K_{0.932}Al_2(Al_{0.932}Si_{3.068}O_{10})((OH)_{1.744}F_{0.256})$ | 7.0 |
| Clinochlore | $Mg_{4.882}Fe_{0.22}Al_{1.88}1Si_{2.96}O_{10}(OH)_8$ | 3.1 |

**Table 2.** *Cont.*

| Albite | $Na(AlSi_3O_8)$ | 1.8 |
|---|---|---|
| Orthoclase | $(K_{0.88}Na_{0.1}Ca_{0.009}Ba_{0.012})(Al_{1.005} S_{2.995}O_8)$ | 1.3 |
| Calcite | $CaCO_3$ | 0.2 |

| **Sayak** | | |
|---|---|---|
| **Compound Name** | **Formula** | **S-Q** |
| Quartz | $SiO_2$ | 20.0 |
| Andradite.aluminian | $Ca_3Al_84Fe_{1.16}Si_3O_{12}$ | 13.5 |
| Wollastonite | $CaSiO_3$ | 11.6 |
| Cronstedite-6 | $Fe_3FeSiO_4(OH)_5$ | 9.6 |
| Cordierite.ferroan.sodian | $Na_{25}(Mg_{1.4}Fe_6)(Al_{3.84}Be_{16})Si_5O_{18}(H_2O)_6$ | 9.6 |
| Donbassite-2Mla | $Al_{4.33}(Si_3Al)O_{10}(OH)_8$ | 8.5 |
| Albite.calcian.ordered | $(Na.Ca)Al(Si·Al)_3O_8$ | 6.6 |
| Calcite | $CaCO_3$ | 5.6 |
| Dolomite | $CaMg(CO_3)_2$ | 3.3 |
| Microcline | $(K_{0.95}Na_{0.5})AlSi_3O_8$ | 2.8 |
| Magnetite syn | $Fe_3O_4$ | 2.6 |
| Muscovite-2M1 | $K_{0.932}Al_2(Al_{0.932}Si_{3.068}O_{10})((OH)_{1.744}F_{0.256})$ | 2.4 |
| Iron Oxide | $Fe_{2.932}O_4$ | 2.1 |
| Clinochlore | $Mg_{2.5}Fe_{1.65}Al_{1.5}Si_{2.2}Al_{1.8}O_{10}(OH)_8$ | 1.7 |

| **Satbayev** | | |
|---|---|---|
| **Compound Name** | **Formula** | **S-Q** |
| Quartz | $SiO_2$ | 54.50 |
| Albite | $Na(AlSi_3O_8)$ | 18.40 |
| Clinochlore-1MIIb (ferroan) | $(Mg.Fe)_6(Si.Al)_4O_{10}(OH)_8$ | 9.20 |
| Potassium sulfite hydrate | $K_2(S_3(SO_3)_2)(H_2O)_{1.5}$ | 6.40 |
| Potassium hydrosulfate | $K_3H(SO_4)_2$ | 2.90 |
| Potassium sulfite hydrate | $K_2(S_3(SO_3)_2)(H_2O)_{1.5}$ | 2.80 |
| Muscovite | $H_2KAl_3Si_3O_{12}$ | 2.40 |
| Hematite (syn) | $Fe_2O_3$ | 1.15 |
| Calcite | $CaCO_3$ | 1.00 |
| Pyrite | $FeS_2$ | 0.80 |
| Malachite | $Cu_2(CO_3)(OH)_2$ | 0.30 |
| Chalcopyrite | $CuFeS_2$ | 0.15 |

Mineralogical analysis showed that copper in the studied ore samples was mainly in the oxidized forms of malachite, as well as in the form of chalcopyrite phenocrysts (Figure 3).

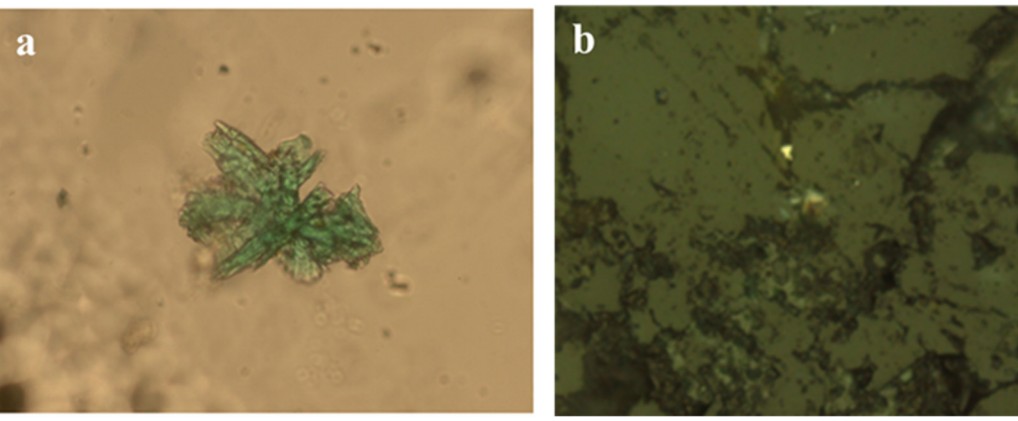

**Figure 3.** Images of mineralogical analysis of the main copper minerals: (**a**) cluster of prismatic malachite crystals (0.5 mm × 0.3 mm); (**b**) chalcopyrite in veinlets of nonmetallic mass (0.04 mm × 0.02 mm).

## 3. Materials and Methods

Experiments intended to extract copper by using sulfuric acid were performed on a large laboratory scale in especially equipped columns—percolators—allowing the heap leaching process to be simulated with the subsequent inclusion of all hydrometallurgical production stages—extraction, re-extraction, and electrowinning sections (Figures 4 and 5).

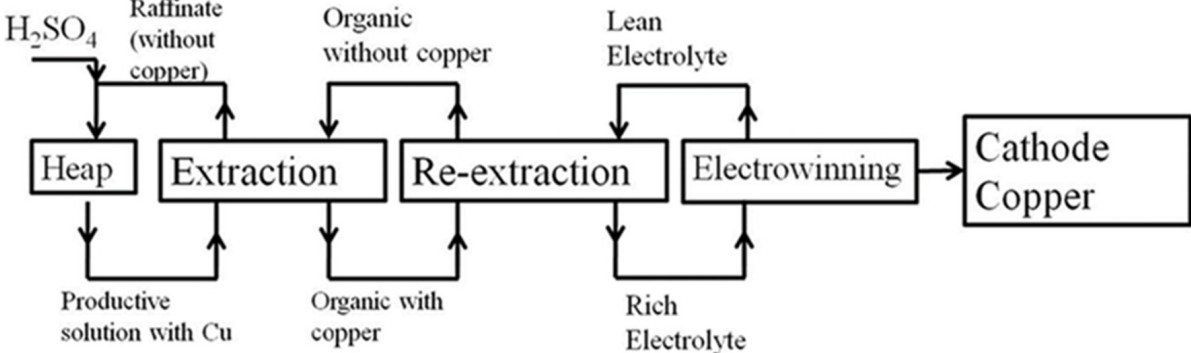

**Figure 4.** General scheme of hydrometallurgical copper production.

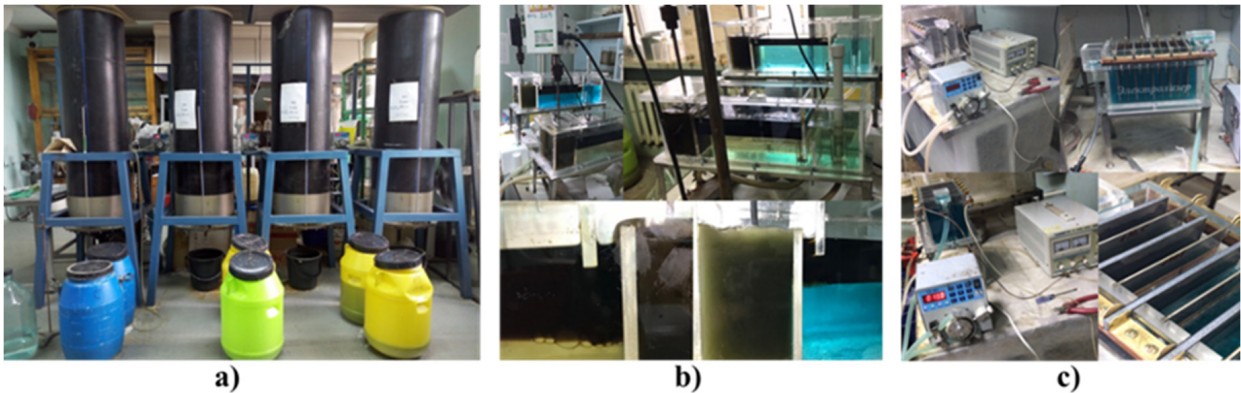

**Figure 5.** Equipment for extensive laboratory tests of hydrometallurgical technology for copper production: (**a**) percolator system; (**b**) extraction and re-extraction section; and (**c**) electrowinning section.

Hydrometallurgical studies of these copper samples from four deposits were conducted over 9 years, from 2014 to 2023. During these studies, the optimal leaching conditions were selected. Lix984 reagent in the form of 10% solution in Escaid solvent was used as an organic extraction phase.

Biochemical oxidation technology with the use of *Acidithiobacillus ferrooxidans* bacterial culture, adapted to the composition of this ore, was also tested on a sample of the Satbayev copper dump in 2022 in addition to the standard sulfuric acid leaching process [18,19]. The adaptation and growth of *A. ferrooxidans* bacterial culture is usually accompanied by certain changes in the solution parameters [20–23]; in particular, there was an active decrease in the $Fe^{2+}$ concentration and an increase in $Fe^{3+}$ ions. Copper compounds are often toxic to the standard strain of *A. ferrooxidans*. Thus, additional microbiological selection with the cultivation of an adapted culture is required. *A. ferrooxidans* strains adapted to the conditions of copper raw materials enable the bioleaching of sulfides, especially copper sulfides [24]. A sample of the *Acidithiobacillus ferrooxidans*-1333 strain, bred in the Korean Centre for Culture Collection, showed high levels of $Fe^{2+}$ oxidation in chalcopyrite due to the characteristic high immobilization of bacteria to this specific mineral [25].

## 4. Experimental Procedure

Large laboratory tests to leach copper were performed after the loading of fixed ore samples into percolators and the installation of extraction sections. The leaching process

was performed using 2.0–2.5% sulfuric acid solutions during the initial period, with a gradual reduction in concentration. The Satbayev dump sample was pretreated using a bacterial solution, with a lower concentration of sulfuric acid during the first 20 days for the biochemical variant of the leaching process, after which the leaching process was performed with 2.5% sulfuric acid solution, initially, followed by a subsequent decrease in concentration. The resulting productive solutions were analyzed daily for copper and acid content, and the amount of sulfuric acid needed to achieve the required concentration was then added; the solution was reused for the leaching process. The product solution was fed to the extraction plant with subsequent leaching through the use of copper-less raffinate. For the calculation of extraction, the values of copper masses removed from the productive solution in the extraction process and the amount of copper in the current solution, containing raffinate, as well as that of residual copper in the organic phase, were summed. The calculation of the total extraction can be expressed by Formula (1) as follows:

$$E_{Cu} = \frac{m(Cu_{pls}) + m(Cu_{raf}) + m(Cu_{el-t}) + m(Cu_{organic})}{m(Cu_{initial})} \cdot 100\% \tag{1}$$

where $m(Cu_{pls})$ is the mass of copper in the productive solution, $m(Cu_{raf})$ is the mass of copper in raffinate after leaching, $m(Cu_{el-t})$ is the mass of copper in electrolyte, and $m(Cu_{organic})$ is the mass of copper in the organic phase. The initial mass of copper $m(Cu_{initial})$ was calculated from the percentage and mass of the ore loaded in the percolator. The copper mass of each phase was calculated with the use of the standard formula (Formula (2)):

$$m(Cu) = C(Cu) \cdot V \tag{2}$$

where $C(Cu)$ is the copper concentration and $V$ is the solution volume. The initial copper mass was calculated from the percentage and mass of the ore loaded in the percolator.

Acid consumption was calculated by dividing the sum of all acid additives by the mass of ore loaded in the percolator, and then recalculated based on kilograms per ton for the current time.

The ratio of acid consumption to the amount of extracted copper was calculated by dividing the sum of all additives by the mass of copper currently extracted from the ore.

The percolation leaching process results, including the calculation of sulfuric acid consumption per ton of ore, the consumption ratio of sulfuric acid and extracted copper, and the final copper recovery from the ore, are presented in Tables 3–7, which show the dynamics of these parameters over a period of 60 days.

**Table 3.** Main parameters of copper recovery and sulfuric acid consumption during the 60-day leaching process for the Satbayev heap.

| Days | Extraction, Cu % | kg $H_2SO_4$ per Ton of Ore | g $H_2SO_4$/g Cu (Extracted into the Solution) |
|---|---|---|---|
| 5 | 6.5 | 6.1 | 35.7 |
| 10 | 14.0 | 8.5 | 25.2 |
| 15 | 23.0 | 10.2 | 24.3 |
| 20 | 34.0 | 11.3 | 19.6 |
| 25 | 45.5 | 13.1 | 9.4 |
| 30 | 58.1 | 14.0 | 9.1 |
| 35 | 68.5 | 14.3 | 8.6 |
| 40 | 75.4 | 14.8 | 7.7 |
| 45 | 81.0 | 15.1 | 7.2 |
| 50 | 82.9 | 15.5 | 7.0 |
| 55 | 84.1 | 15.5 | 6.9 |
| 60 | 86.7 | 15.5 | 6.8 |

**Table 4.** Main parameters of copper recovery and consumption of sulfuric acid during the 60-day leaching process for the Satbayev heap (bioleaching).

| Days | Extraction, Cu % | kg $H_2SO_4$ per Ton of Ore | g $H_2SO_4$/g Cu (Extracted into the Solution) |
|---|---|---|---|
| 5 | 0.4 | 1.5 | 140.0 |
| 10 | 2.1 | 4.0 | 61.6 |
| 15 | 11.3 | 6.0 | 20.2 |
| 20 | 18.0 | 6.5 | 20.2 |
| 25 | 25.0 | 8.1 | 13.0 |
| 30 | 33.7 | 8.8 | 9.9 |
| 35 | 42.4 | 9.2 | 8.3 |
| 40 | 55.4 | 9.2 | 6.3 |
| 45 | 64.8 | 9.4 | 5.4 |
| 50 | 71.3 | 9.4 | 5.1 |
| 55 | 79.3 | 9.4 | 4.7 |
| 60 | 87.2 | 9.4 | 4.1 |

**Table 5.** Main parameters of copper recovery and sulfuric acid consumption during the 60-day leaching process for Bayskoye.

| Days | Extraction, Cu % | kg $H_2SO_4$ per Ton of Ore | g $H_2SO_4$/g Cu (Extracted into the Solution) |
|---|---|---|---|
| 5 | 11.83 | 12.44 | 27.9 |
| 10 | 27.84 | 18.7 | 17.8 |
| 15 | 34.1 | 19.9 | 15.5 |
| 20 | 38.5 | 20.32 | 14.0 |
| 25 | 43.5 | 20.91 | 12.8 |
| 30 | 46.2 | 21.08 | 12.1 |
| 35 | 50.1 | 21.15 | 11.2 |
| 40 | 56.89 | 21.66 | 10.1 |
| 45 | 59.6 | 22.24 | 9.9 |
| 50 | 64.32 | 22.31 | 9.2 |
| 55 | 66.0 | 22.64 | 9.1 |
| 60 | 66.13 | 22.69 | 9.1 |

**Table 6.** Main parameters of copper recovery and sulfuric acid consumption during the 60-day leaching process for Baitemir.

| Days | Extraction, Cu % | kg $H_2SO_4$ per Ton of Ore | g $H_2SO_4$/g Cu (Extracted into the Solution) |
|---|---|---|---|
| 5 | 10.1 | 9.23 | 14.2 |
| 10 | 14.6 | 9.26 | 9.85 |
| 15 | 18.1 | 9.33 | 7.15 |
| 20 | 22.3 | 9.49 | 6.33 |
| 25 | 26.2 | 9.57 | 5.44 |
| 30 | 30.3 | 10.22 | 5.24 |
| 35 | 34.9 | 11.46 | 5.1 |
| 40 | 38.5 | 13.88 | 5.6 |
| 45 | 44.5 | 14.81 | 5.17 |
| 50 | 48.3 | 15.54 | 5.0 |
| 55 | 51.8 | 15.67 | 4.7 |
| 60 | 54.3 | 15.83 | 4.53 |

**Table 7.** Main parameters of copper recovery and sulfuric acid consumption during the 60-day leaching process for Sayak.

| Days | Extraction, Cu % | kg $H_2SO_4$ per Ton of Ore | g $H_2SO_4$/g Cu (Extracted into the Solution) |
|---|---|---|---|
| 5 | 15.67 | 35.61 | 94.68 |
| 10 | 20.81 | 39.72 | 79.52 |
| 15 | 26.07 | 43.97 | 70.28 |
| 20 | 32.41 | 45.36 | 58.32 |
| 25 | 35.6 | 46.27 | 54.15 |
| 30 | 38.2 | 46.97 | 51.23 |
| 35 | 45.5 | 51.61 | 47.26 |
| 40 | 60.5 | 64.41 | 44.36 |
| 45 | 74.5 | 65.98 | 36.9 |
| 50 | 77.9 | 66.28 | 35.45 |
| 55 | 78.4 | 66.36 | 35.27 |
| 60 | 78.6 | 66.4 | 35.2 |

When the percolation leaching process was completed, samples of ore materials were analyzed for residual copper content and forms (Table 8).

**Table 8.** Residual copper in ore samples after the leaching process (%).

| Satbayev | Satbayev (Bioleaching) | Bayskoye | Baytemir | Sayak |
|---|---|---|---|---|
| 0.036 | 0.033 | 0.13 | 0.31 | 0.055 |

From the mineralogical analysis, it was found that the remaining copper in all samples was mainly represented by the complex oxide $K_2Cu_2Mg_3Si_{12}O_{30}$ and other complexes of silicate phenocrysts (Figure 6).

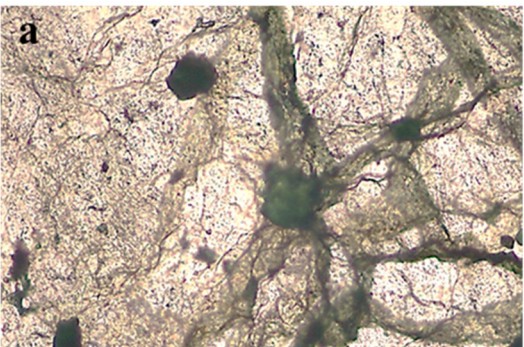 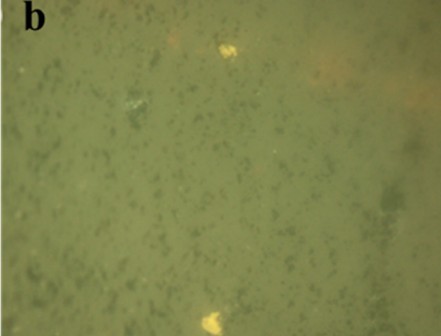

**Figure 6.** Mineralogical analysis images of residual copper compounds after the leaching process: (**a**) complex oxide $K_2Cu_2Mg_3Si_{12}O_{30}$; (**b**) silicate phenocrysts of copper.

A detailed study of residual copper phenocrysts was performed with the use of a JXA-8230 electron probe microanalyzer manufactured by JEOL. This provided the corresponding spectra for oxygen, silicon, and aluminum, which are characteristic of aluminosilicates, and evidence of their presence in these particles (Figure 7).

These fragments were not found in the original samples, and nor was the detected complex oxide. This fact indicates that, like calcite, as mentioned earlier [9], some silicate compounds can act as sorbents for dissolved copper ions. At the same time, oxidative processes in several similar mineral fragments can proceed, resulting in the observed geological changes in deposit rock, as was established by [26], which will allow a for more complete extraction of copper after some time.

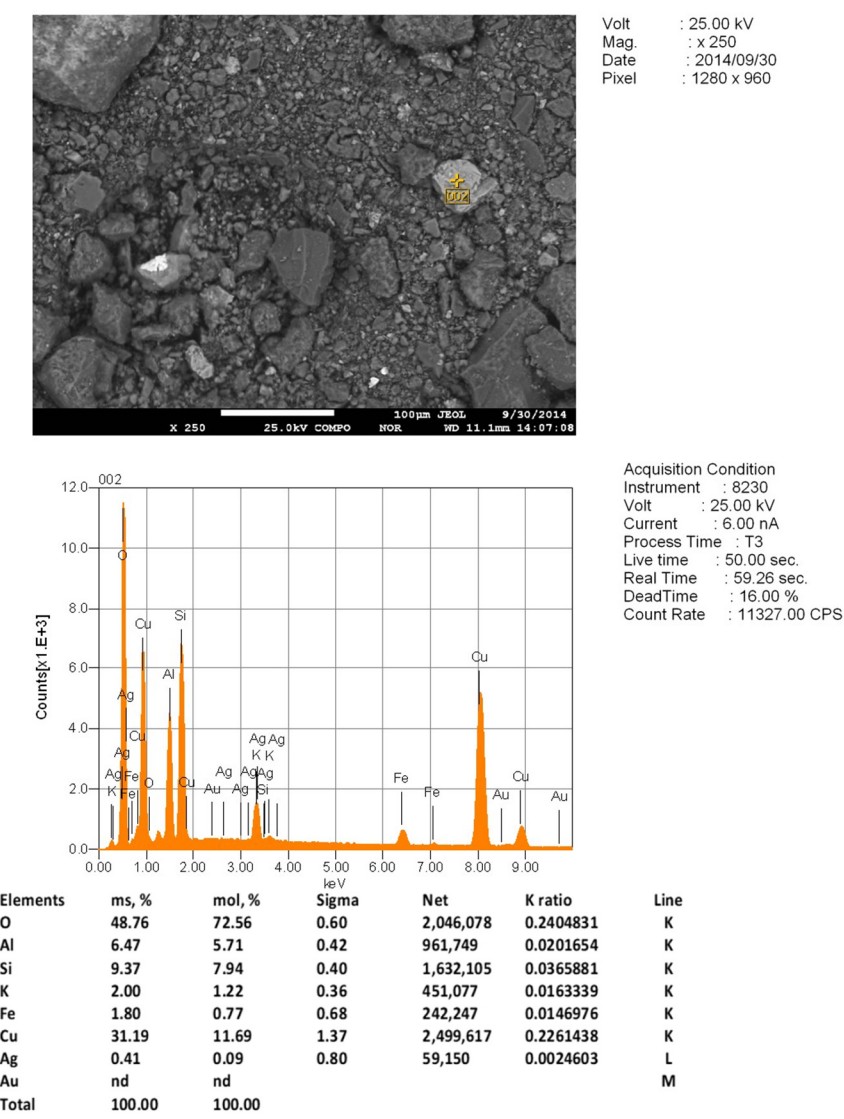

**Figure 7.** Electron microscopy analysis of disseminated residual copper particles.

## 5. Discussion of the Results

The results of the percolation leaching experiments showed that the most rapid increase in copper recovery occurred in the first ten days for the Sayak and Bayskoye ore samples. At the same time, a significant amount of sulfuric acid addition was needed to achieve stabilized pH values and the required concentration of sulfuric acid. For samples from Baitemir and Satbayev deposits, less sulfuric acid was required to stabilize the acidic environment, while there was no sharp increase in the copper concentration in productive solutions, as in other samples.

In the biochemical leaching variant of the Satbayev sample, the bacterial culture was adapted with an accompanying bio-oxidation of the mineral raw materials at pH levels of 2.0–2.5 at the initial stage, and the sulfuric acid concentration was not increased more than 0.5%. Therefore, during the initial twenty days, the amount of extracted copper in the solution remained at a rather low level, and the mass ratio to the extracted metal was extremely high, even at a small consumption rate of acid per ton of ore. Productive solutions for biochemical leaching processes were not subjected to extraction processing in the initial period.

Over 60 days, the amount of additive required to maintain the appropriate concentration of sulfuric acid steadily decreased, resulting in a decrease in the mass consumption ratio of acid to recoverable copper. No further addition of acid to the circulating leaching

solution of raffinate after extraction was needed for the standard and biochemical leaching samples of the Satbayev deposit after 45–50 days of the process, because the acidity parameters of the medium at that point corresponded to the required parameters ($H_2SO_4$: 2.0–2.5%; pH: 1.5–2.0).

The results of copper extraction on the solution and the sulfuric acid consumption level obtained for 60 days of the leaching process had different values for each deposit and were dependent on the initial content in the ore. The graphs in Figure 8 show a comparison of the sulfuric acid leaching results of the four deposits, with an additional option of the biochemical leaching process. Thus, despite the lowest result of 54.3% extraction in the sample of Baitemir ore with an initial copper content of 0.64%, the weight of the extracted copper in this case was 545 g. Only 377 g of metal was extracted in the solution from the Sayak deposit sample, with an initial copper content of 0.24% corresponding to an extraction of 78.6%. Approximately the same amount of copper was extracted in the solutions during the leaching process of the Bayskoye deposit samples (514 g, Satbayev; 515 and 518 g (bio-option)), but the extraction level also differed significantly considering the differences in the starting copper content in these samples.

If the consumed acid balance was recalculated according to the total amount of extracted copper, the highest efficiency was found in the experiment with preliminary biooxidation of ore from the Satbayev dump; $H_2SO_4/Cu = 1{:}4.1$, whereas this ratio was 1:6.8 during the standard leaching process. The Baitemir ore sample leaching process showed a final ratio of $H_2SO_4/Cu = 1{:}4.53$ despite the extraction rate of 54.3%. The balance of acid consumption and extracted metal was 1:9.1 in the Bayskoye ore sample. The highest acid consumption was observed in the Sayak sample, where the final mass ratio to the metal was $H_2SO_4/Cu = 1{:}35.2$.

The sulfuric acid consumption level per ton of ore depended largely on the initial content of copper and its extraction efficiency in the solution. According to the calculation of acid consumption in the leaching process based on the recalculation per ton of ore, the lowest values were found for the Satbayev deposit samples; 15.5 kg per ton of ore, and 9.4 kg per ton of ore in the case of bacterial oxidation. The consumption of acid per ton of ore was 15.83 kg in experiments with Baitemir ore, and 22.69 kg with Bayskoye ore. The highest consumption in this ratio was found in the leaching of the Sayak sample, which was 66.4 kg of acid per ton of ore.

The calculation of these parameters enables one to estimate the amount of sulfuric acid that will be required for the hydrometallurgical processing of copper waste and, consequently, to estimate the profitability of copper cathode production at a particular deposit or site.

The results obtained for sulfuric acid consumption rates were compared with the initial data on the content of calcium and magnesium compounds in the studied copper deposit samples. Thus, Table 9 shows the total contents of the alkaline earth metals calcium and magnesium; the contents of their compounds in the form of calcite, dolomite, silicates, and sulfates (gypsum, basanite); and the acid consumption level for each ore sample.

**Table 9.** Content of alkaline earth metals and their compounds in copper ore samples (%); sulfuric acid consumption level.

| Sample | $Ca^{2+}$ and $Mg^{2+}$ | Calcium and Magnesium Compounds | | | Total Number of Ca and Mg Compounds | Mass Ratio of Acid/Cu in Solution | kg of Acid per Ton of Ore |
|---|---|---|---|---|---|---|---|
| | | Calcites and Dolomites | Silicates of Ca and Mg | Gypsum and Basanite | | | |
| Satbayev | | | | | | 6.8 | 15.5 |
| Satbayev (bioleaching) | 3.8 | 1.0 | 9.2 | <0.05 | 10.2 | 4.1 | 9.4 |
| Bayskoye | 5.0 | 1.5 | 9.7 | 6.9 | 18.1 | 9.1 | 22.69 |
| Baitemir | 7.48 | <0.2 | 14.9 | 10.6 | 25.7 | 4.53 | 15.83 |
| Sayak | 10.16 | 8.9 | 26.8 | <0.05 | 35.7 | 35.2 | 66.4 |

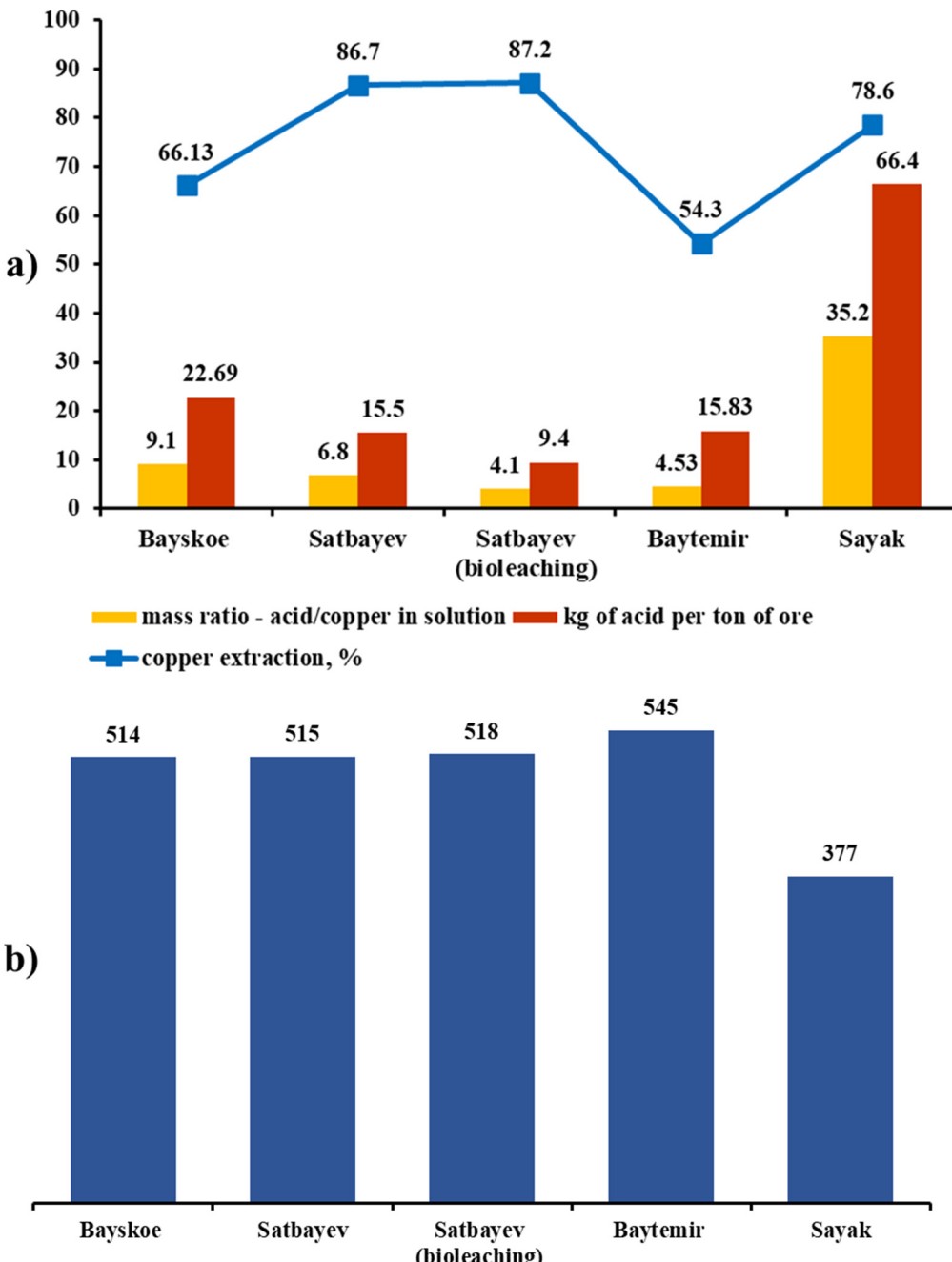

**Figure 8.** Comparison of final parameters of acid consumption (per ton of ore and mass ratio to copper in solution) and extraction of copper from ore ((**a**)—in %, (**b**)—in grams).

The comparison of the acid consumption results for the four different copper deposits in the leaching process showed that silicate compounds of calcium and magnesium, in addition to calcite and dolomite, also affected the increase in acid consumption. At the same time, calcium sulfate compounds of gypsum and basanite types found in samples from Bayskoye and Baitemir deposits did not affect the acid consumption during the leaching process. The graphs in Figures 9 and 10 show comparisons of acid consumption during the leaching process (including one option of bioleaching) of copper deposit samples with different contents of calcium and magnesium compounds based on data in Table 9.

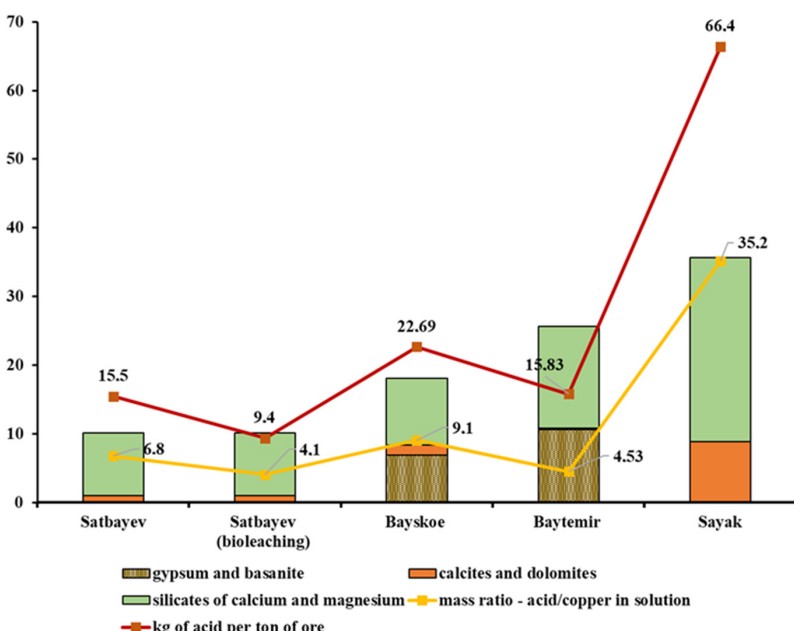

**Figure 9.** Effect of calcium and magnesium compounds on the level of sulfuric acid consumption during the leaching process.

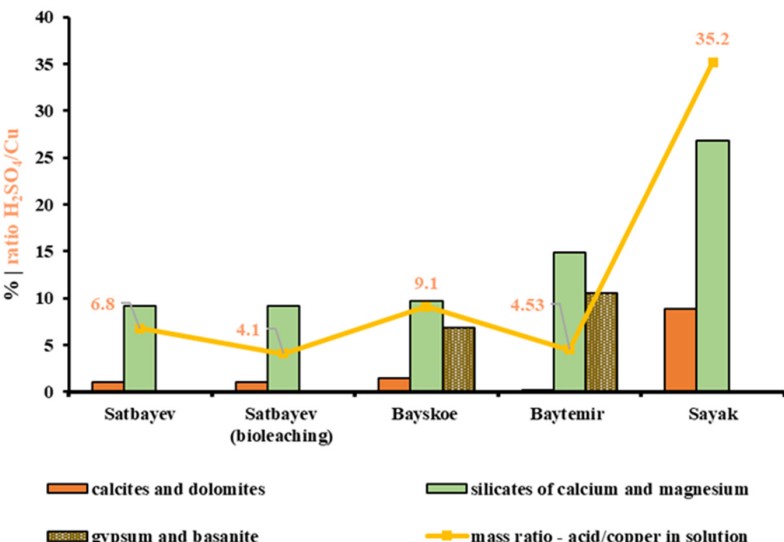

**Figure 10.** Types of calcium and magnesium compounds and their effect on the mass ratio of consumed sulfuric acid to extracted copper.

The highest consumption of sulfuric acid during the leaching process for the Sayak deposit sample occurred because of the high content of calcium- and magnesium-containing minerals in the ore, with the total mass fraction of 35.7%. The most actively absorbing acid minerals, such as calcite and dolomite, accounted for 8.9%; the remainder, namely, 26.8%, was composed of calcium and magnesium silicates.

In the Baitemir ore sample, the level of sulfuric acid consumption during the leaching process was relatively low despite the rather high total content of calcium and magnesium minerals, namely, 25.7%. This factor was caused by the extremely low content of calcite and dolomite in the sample; not more than 0.2%. Silicates of alkaline earth metals less actively absorbing sulfuric acid made up 14.9%; the rest, namely, 10.6%, consisted of gypsum and basanite, which are inert to sulfuric acid.

A different result was found for the Bayskoye deposit sample. The total consumption of acid per ton of ore was 22.69 kg at a total calcium and magnesium mineral content

of 18.1%, of which sulfates in the form of gypsum and basanite accounted for 6.9%. In this case, a significant consumption was caused by the combination of compounds of the alkaline earth minerals carbonate (1.5%) and silicate (9.7%).

The sulfuric acid consumption was 15.5 kg per ton for the Satbayev deposit sample, with a total content of alkaline earth minerals of 10.2%, according to the results of the standard leaching process, which was comparable with the Baitemir sample results. However, considering the difference in the initial copper content in these deposits, the balance of the mass ratio of spent acid to recovered copper in the standard leaching was $H_2SO_4/Cu = 6.8:1$, which was one-third higher than the corresponding result for the leaching of Baitemir ore.

The biochemical leaching experiment for the Satbayev deposit sample deserves special attention. The amount of sulfuric acid finally consumed per ton of ore was 9.4 kg, and on the recalculation of the recovered copper balance the ratio was $H_2SO_4/Cu = 4.1:1$. This effect of the decreased sulfuric acid consumption was due to the partial regeneration of sulfuric acid during the oxidation of sulfide fragments present as part of the ore composition.

*5.1. Mechanism of Oxidative Reactions*

The practice of using oxidizing reagents is widespread in the field of hydrometallurgical gold production. Methods using peroxides, chloroactive compounds, surfactants, and bacterial culture applications for the decomposition of host gold sulfides such as pyrite, arsenopyrite, etc., are well known [27–29]. However, in principle, oxidizing factors play a different and contrasting role in gold-bearing minerals in oxidation during the leaching process for copper deposits. Moreover, the use of several oxidizing factors, especially those containing active chlorine, is impossible and unprofitable for the liquid extraction technology used for copper production.

The biotechnological copper leaching process involves the irrigation of ore materials or technogenic waste-containing metal sulfides using sulfuric acid solutions and iron salts, as well as the introduction of viable thionic, iron-oxidizing bacteria. Conventional oxidation of the sulfide minerals most found in copper dumps, with the use of pyrite and chalcopyrite as examples, can be described by the following reactions (3,4):

$$2FeS_2 + 7O_2 + 2H_2O \rightarrow 2FeSO_4 + 2H_2SO_4 \tag{3}$$

$$CuFeS_2 + 4O_2 \rightarrow CuSO_4 + FeSO_4 \tag{4}$$

However, the presence of iron (II) and other metals that are less active than copper in solution may contribute to its precipitation if they are not present in the maximum oxidation degree, as exemplified by the following reaction (5):

$$Cu^{2+} + 2Fe^{2+} \rightarrow Cu^0\downarrow + 2Fe^{3+} \tag{5}$$

Oxidative processes in sulfuric acid media, with the participation of air oxygen, will convert iron compounds from +2 to +3 oxidation degrees according to the following reaction (6):

$$4FeSO_4 + O_2 + 2H_2SO_4 \rightarrow 2H_2O + 2Fe_2(SO_4)_3 \tag{6}$$

During the leaching process, the presence of a catalyzing factor in sulfuric acid media helps to accelerate the transition of iron to the +3 oxidized form, as exemplified in the following reactions (7,8):

$$2FeS_2 + 14H_2SO_4 \rightarrow 14H_2O + Fe_2(SO_4)_3 + 15SO_2 \tag{7}$$

$$2CuFeS_2 + 18H_2SO_4 \rightarrow 18H_2O + Fe_2(SO_4)_3 + 17SO_2 + 2CuSO_4 \tag{8}$$

The obtained iron (III) compounds can act as oxidizers. The transfer of iron ions with the maximum oxidation degree of +3 into the productive solution contributes to the further oxidation of acid-absorbing and noble copper-containing minerals during the circulation of the productive solution after the extraction stage (leaching with raffinate).

Regarding practical usage, iron (III) sulfate is known as an oxidizing catalyst. In the leaching process, it can contribute to the dissolution of sulfide minerals of copper through the following reactions (9,10):

$$CuFeS_2 + 2Fe_2(SO_4)_3 + 2H_2O + 3O_2 \rightarrow CuSO_4 + 2H_2SO_4 + 5FeSO_4 \tag{9}$$

$$2CuS + 2Fe_2(SO_4)_3 + 2H_2O + O_2 \rightarrow 2CuSO_4 + 2H_2SO_4 + 4FeSO_4 \tag{10}$$

The practice of using trivalent iron sulfate as an oxidizing agent is widespread for copper deposits and in underground uranium leaching processes [30]. At the same time, hydrometallurgical facilities that develop copper deposits with high iron contents face the widespread problem of excessive accumulation, and therefore concentrations of iron ions in the productive solution. The excessive concentration of trivalent iron ions of more than 10 g/L negatively affects the extraction process because it reduces the selectivity of the organic extractant for copper, which deteriorates the quality of the electrolyte and copper cathode during electrowinning. It is inexpedient to consider an iron (III) sulfate additive as an oxidizing reagent in such cases, wherein the cultivation of iron-oxidizing microorganisms will allow an optimum concentration of +3 iron ions to be obtained from the initial content in the ore material.

### 5.2. Assessment of Economic Efficiency

The results of acid consumption studies facilitate the assessment of the cost-effectiveness of processing a particular copper deposit when considering the dynamics of copper prices and the cost of sulfuric acid. The cost of the main reagent, sulfuric acid, largely depends on the mining region, the location of the nearest sulfuric acid plant, the organization of logistics, and transportation to the deposit. Thus, the minimum price for a sulfuric acid ton may be USD 40 per ton, while at the same time, in some countries, the price per ton may reach USD 280 [31]. According to the London Metal Exchange (LME) [32], the cost per ton of copper over the past three years (2020–2023) ranged from USD 5500 to USD 10,000. The profit per ton of copper produced is calculated based on these price factors and taking into account the additional production costs, which are about 25% on average. These results are presented in Table 10.

**Table 10.** Calculation of net profit per ton of copper produced, in USD.

| Ore Deposit | Mass Ratio of Acid/Cu in Solution | Price per Ton of Copper | Price per Ton of Sulfuric Acid | | |
| --- | --- | --- | --- | --- | --- |
| | | | Minimum USD 40/t | Average USD 160/t | Maximum USD 280/t |
| Satbayev | 6.8 | Maximum USD 10,000/t | 7228 | 6412 | 5596 |
| | | Average USD 7500/t | 5353 | 4537 | 3721 |
| | | Minimum USD 5500/t | 3853 | 3037 | 2221 |
| Satbayev (bioleaching) | 4.1 | Maximum USD 10,000/t | 7336 | 6844 | 6352 |
| | | Average USD 7500/t | 5461 | 4969 | 4477 |
| | | Minimum USD 5500/t | 3961 | 3469 | 2977 |
| Bayskoye | 9.1 | Maximum USD 10,000/t | 7136 | 6044 | 4952 |
| | | Average USD 7500/t | 5261 | 4169 | 3077 |
| | | Minimum USD 5500/t | 3761 | 2669 | 1577 |
| Baytemir | 4.53 | Maximum USD 10,000/t | 7319 | 6775 | 6231 |
| | | Average USD 7500/t | 5444 | 4900 | 4356 |
| | | Minimum USD 5500/t | 3943 | 3400 | 2856 |
| Sayak | 35.2 | Maximum USD 10,000/t | 6092 | 1868 | −2356 |
| | | Average USD 7500/t | 4217 | −7 | −4231 |
| | | Minimum USD 5500/t | 2717 | −1507 | −5731 |
| **High profit** | | **Low profit** | | | **Losses** |

The table shows that most of the copper deposits are economically profitable, and their processing will be profitable even at the maximum price fluctuations of the considered range. There would also be an increase in net profit with the use of the biochemical leaching process for the Satbayev deposit ore.

A different picture was observed in the economic assessment of the Sayak deposit ore. Thus, taking into account the high consumption of sulfuric acid for recoverable copper, the economic feasibility of hydrometallurgical processing of ores of this deposit is possible only if there is a source of cheap sulfuric acid.

## 6. Conclusions

In conclusion, the dependence of the sulfuric acid consumption rate on the amount and forms of alkaline earth metal mineral content was established following a comparison of results obtained during sulfuric acid leaching experiments conducted from 2014 to 2023 at four different copper deposits. In addition to the known fact about the ability of alkaline earth metal carbonate minerals to neutralize an acidic environment, some calcium and magnesium silicates were also found to increase sulfuric acid consumption in the leaching process. Minimum levels of sulfuric acid consumption were noted during the leaching process for samples from Satbayev and Baitemir deposits, and maximum consumption levels were found in experiments with Sayak ore ($H_2SO_4/Cu = 35.2:1$).

The analysis of the ore material after the experiments showed that copper was present mainly in the form of complex silicate complexes in all samples, which were not found in the original samples. This was due to the sorption ability of silicates of alkaline earth metals and the accumulation of copper ions reacting with sulfate compounds in solution.

The high efficiency of copper extraction from the Satbayev deposit was due to the relatively low content of calcium and magnesium compounds in the ore compared with samples from other deposits. At the same time, in the sample of the Baytemir deposit, despite the relatively low extraction of 54.3% in 60 days, there was a tendency for the ratio of sulfuric acid consumption to extracted copper to decrease; this was due to the high reserve in the copper ore and the presence of calcium and magnesium compounds in a less acid-absorbing form relative to the remaining samples. The extraction of copper from the Bayskoye deposit for the same period was 66.13%, with a moderate consumption of sulfuric acid due to the presence of a noticeable amount of carbonate and silicate compounds of calcium and magnesium in the ore.

The reverse situation was observed in the sample from the Sayak deposit. With a sufficiently high extraction of 78.6%, the maximum amount of sulfuric acid was required to achieve and maintain the necessary parameters for the leaching solution. This deposit also had the lowest copper content of the studied deposits, which was reflected in the extremely high ratio of sulfuric acid consumption to the amount of extracted metal. The highest consumption of sulfuric acid during the leaching of the Sayak deposit sample was due to the high content of calcium- and magnesium-containing minerals in the ore rock, the total mass fraction of which was 35.7%. The most actively acid-absorbing minerals, calcite and dolomite, accounted for 8.9%, while the remaining 26.8% was represented by calcium and magnesium silicates.

The bacterial oxidation experiment of the Satbayev ore sample conducted in parallel with the standard leaching process showed the possibility of significantly reducing sulfuric acid consumption while achieving a sufficiently high level of copper extraction of 87.2%. This effect was achieved due to oxidative mechanisms occurring in the process of the by-product decomposition of minerals containing sulfur and iron. Sulfide minerals (pyrite and chalcopyrite) present in small quantities in the Satbayev ore sample after bacterial oxidation treatment facilitate the partial recovery of sulfuric acid, which eventually affects the overall consumption level.

The achieved effect of reduced sulfuric acid consumption was especially important given the high cost of this reagent. Thus, according to cost-effectiveness calculations under the maximum variant of sulfuric acid price, the use of the biochemical leaching method for

the Satbayev deposit would yield an additional USD 700 for each ton of copper produced compared to the standard leaching method. The Bayskoye and Baitemir copper deposits also have economic potential and can generate profits even under adverse price conditions in the sulfuric acid and metal exchange markets. Preliminary economic calculations for the Sayak mine showed that the processing of such a deposit is lucrative only if there is an inexpensive supply of sulfuric acid or the organization itself produces this reagent.

**Author Contributions:** A.K.: Project administration, Methodology, Validation; B.K.: Supervision, Conceptualization, Writing—Review & Editing; D.M.: Investigation, Writing—Original Draft, Formal analysis; E.K.: Resources, Investigation; M.Y.: Formal analysis, Methodology; A.B.: Conceptualization, Investigation; N.A.: Investigation, Resources. All authors have read and agreed to the published version of the manuscript.

**Funding:** This research was supported by a grant project of the Science Committee of the Ministry of Education and Science of Republic of Kazakhstan, project No. AP09258789.

**Data Availability Statement:** https://imio.kz/en/laboratory-of-special-methods-of-hydrometallurgy/; https://imio.kz/en/list-of-innovative-projects/ (accessed on 2 May 2023).

**Conflicts of Interest:** The authors declare no conflict of interest.

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
