# Peer review of "Study of Factors Affecting the Copper Ore Leaching Process"

_2305-7084, doi:10.3390/chemengineering7030054_

Round 1

Reviewer 1 Report

1. The image dimensions are missing in Figure 3.

2. The paper has done a large number of long-term experiments on ore leaching. However, the significance of this study is not clear, and the introduction of the problem to be solved is not clear.

3. In the article, it is necessary to explain the comparative content of the four minerals. In particular, the differences between the four minerals should be analyzed in the discussion. Otherwise, it is not necessary to use four types of ores for testing.

Author Response

  1. (a) - cluster of prismatic malachite crystals (0.5Ñ…0.3 mm); (b) - chalcopyrite in veinlets of nonmetallic mass (0.04Ñ…0.02 mm)

  1. The article presents studies of the problem of processing off-balance sheet copper deposits accumulated in large quantities in the form of dumps. The presented results demonstrate the limiting possibilities of copper extraction and reflect the level of sulfuric acid consumption, depending on the initial composition of mineral raw materials on the example of four deposit options. Based on these results, the conditions of profitability of processing of a particular deposit were assessed. Using the example of the Satbayev deposit, which contains minerals containing divalent iron, the possibility of effective application of bacterial treatment, which significantly reduces the consumption of sulfuric acid and increases the profitability of copper production, is shown.

  1. In the course of copper leaching experiments, the regularity of increasing the consumption of sulfuric acid from the content of calcium and magnesium minerals was revealed. It was found that in addition to acid-absorbing calcite and dolomite, calcium silicates influence the increase in acid consumption. The pyrite and chalcopyrite minerals found in one of the four deposits made it possible to carry out bacterial oxidation of this ore, which in turn contributed to a decrease in the level of sulfuric acid consumption, due to the mechanism of partial regeneration.

We also processed the article in the English Editing

Reviewer 2 Report

The article cannot be published in ChemEngineering in its present form.

The paper submitted for peer review is more of an introduction to a research paper than an overview. Firstly, in a full-fledged review article, the number of sources considered should be at least 100, while in this paper only 30 references are presented, including links to Internet sources, which is undesirable. Secondly, the article is based on a narrowly focused approach - only the results of sulfuric acid leaching of copper from the deposits of Central Kazakhstan are considered. For an international journal with a wide readership, the information presented is of extremely little interest. At the same time, the problem of extracting such an important metal is very relevant all over the world. It is necessary to expand the list of literature (bring it in line with the world level) and consider modern methods of extracting copper, such as flotation processes, extraction processes using diaminopyridines, organophosphorus extractants, ionic liquids, copper sorption by magnetic polymeric sorbents, their copper extraction efficiency, it is desirable to compare them with natural sorbents (minerals). The current trends in the processing of copper sulfide ores should be assessed. Otherwise, the review seems very poor and uninteresting. The study presented by the authors can be used as the head of a full review.

It should also be clarified what is the reason for the higher efficiency of copper extraction (in %) after 60 days from the Satbaevskoye deposit compared to the Baiskoye, Baitemirskoye and Sayakskoye deposits? In the section Discussion of the results, it is necessary to provide an analysis of the obtained data on the extraction of copper.

Author Response

We agree with the comment about the review article and ask you to consider this work as a regular research article.

The high efficiency of copper extraction from the Satbayev deposit is due to the relatively low content of calcium and magnesium compounds in the ore compared to samples from other deposits. At the same time, in the sample of the Baytemir deposit, despite the relatively low extraction of 54.3% in 60 days, there was a tendency to decrease the ratio of sulfuric acid consumption to extracted copper, this is due, relative to the remaining samples, to a high reserve in the copper ore and the presence of calcium and magnesium compounds in a less acid-absorbing form.  The extraction of copper from the Bayskoye deposit for the same period amounted to 66.13%, with a moderate consumption of sulfuric acid, due to the presence of a noticeable amount of carbonate and silicate compounds of calcium and magnesium in the ore.

The reverse situation was observed in the sample of the Sayak deposit – with a sufficiently high extraction of 78.6%, the use of the maximum amount of sulfuric acid was required to achieve and maintain the necessary parameters of the leaching solution. Taking into account the lowest copper content compared to other deposits, this was reflected in an extremely high ratio of sulfuric acid consumption to the amount of extracted metal. The greatest consumption of sulfuric acid during the leaching of the Sayak deposit sample was due to the high content of calcium and magnesium-containing minerals in the ore rock, the total mass fraction of which was 35.7%. The most actively acid-absorbing minerals calcite, such as calcite and dolomite, accounted for 8.9%, the remaining 26.8% were calcium and magnesium silicates.

In addition, in the case of the Satbayev deposit, compounds containing ferrous iron were found in the mineral raw materials, which made it possible to implement preliminary bacterial oxidation. The use of bacterial culture Acidithiobacillus Ferrooxidans, intensifies oxidative processes, the mechanism of which promotes partial regeneration of sulfuric acid.

We also processed the article in the English Editing

Reviewer 3 Report

Koizhanova et al. studied the factors affecting the copper ore leaching process by analyzing data from the use of liquid-extraction technology at four copper deposits with varying compositions. Based on their findings, it is recommended that the current manuscript be resubmitted as a research paper rather than a review paper, given that it contains significant experimental data and results as well as discussion sections.

Author Response

Yes, we agree with the recommendation and are ready to submit this manuscript as a research article.

We also processed the article in the English Editing

Reviewer 4 Report

The published overview of hydrometallurgical copper extraction study is interesting paper which summarize informácie o 4 typoch Cu-depozites.

The manuscript needs revision.

1. Explain the meaning of variables in equation (1). Line 155 could follows after the Equation (1).

2. Tables 3-7 duplicate the information given by Figure 6. They could be omitted.

3. Figures are adequate, but clarity is missing in most of them; the legends of Fig. 8 and Fig.9 is hard to read. In Fig. 10,  x-axe is missing. The different styles of Fig. 10 and 11 is obvious.

4. What is the meaning of colored rows and columns in Table 10? The explanation into the text is necessary.

5. References are stated chaotically, inconsistently. A uniform layout is required according to manuscript rules.

Author Response

We tried to correct all the comments. It is unclear the remark on literary references - they were made using a template.

We also processed the article in the English Editing

Round 2

Reviewer 2 Report

The article may be published in the journal ChemEngineering after minor corrections.

1. It is necessary to change the phrase about the review article (lines 60-61), since the article is not a review article.

2. Should be improved the Introduction by adding links to modern articles on the extraction processing of ore solutions (see, for example, works https://doi.org/10.3390/chemengineering6010006, https://doi.org/10.3390/met12081275, https:/ /doi.org/10.3390/gels8080492).

3. It is desirable to provide data on the effect of pH, T, time and stirring speed on the copper leaching process (dependence graphs of metal leaching, % =f(time), metal leaching, % =f(pH), metal leaching, % = f(stirring speed, rpm).

4. Correct the design of the list of references according to the requirements of the journal ChemEngineering.

Author Response

1. It is necessary to change the phrase about the review article (lines 60-61), since the article is not a review article.

- Replaced "review" to "article"

2. Should be improved the Introduction by adding links to modern articles on the extraction processing of ore solutions (see, for example, works https://doi.org/10.3390/chemengineering6010006, https://doi.org/10.3390/met12081275, https:/ /doi.org/10.3390/gels8080492).

-Added links

3. It is desirable to provide data on the effect of pH, T, time and stirring speed on the copper leaching process (dependence graphs of metal leaching, % =f(time), metal leaching, % =f(pH), metal leaching, % = f(stirring speed, rpm).

- In our studies, leaching was carried out by percolation. The pH parameters were almost stable 1.5-2.0. For bio-oxidation at the initial stage 2.0-2.5 (The data is in the text). 
Initially, the article included a graph of the dynamics of copper extraction for 60 days, but we were also told on the review that only tables were enough.

4. Correct the design of the list of references according to the requirements of the journal Chem Engineering.

-The incorrect numbering has been restored, as well as the style of links in the text.

Reviewer 3 Report

Publish as it is. 

Author Response

Minor amendments were made to the article and 3 links were added
